# Simulation of Evacuation from Stadiums and Entertainment Arenas of Different Epochs on the Example of the Roman Colosseum and the Gazprom Arena

Marina Gravit [1], Ekaterina Kirik [2], Egor Savchenko [3], Tatiana Vitova [2] and Daria Shabunina [1,*]

1 Peter the Great St. Petersburg Polytechnic University, 195251 St. Petersburg, Russia; marina.gravit@mail.ru
2 Institute of Computational Modelling of the Siberian Branch of the Russian Academy of Sciences, 660036 Krasnoyarsk, Russia; kirik@icm.krasn.ru (E.K.); vitova@icm.krasn.ru (T.V.)
3 Design and Research Institute of Air Transport Lenaeroproekt, 198095 St. Petersburg, Russia; e.savchenko@fakt-group.ru
* Correspondence: shabunina.de@edu.spbstu.ru

**Abstract:** Space-planning decisions of two sports and entertainment arenas with large crowds—the Roman Colosseum (Italy) and the modern Gazprom Arena stadium (St. Petersburg, Russia)— were analyzed to compare the flow of people during evacuation by simulation. It was shown that the space-planning decisions of the Colosseum seem more advantageous compared with the Gazprom Arena in calculation of evacuation time and evacuation organization process: the capacity of the stairs of the Colosseum with a width of 2.8 m is comparable with the capacity of the Gazprom Arena's stairs (4 m). In the Colosseum the average specific flow is $q_{average}$ = 1.14 person/s/m, while in the Gazprom Arena the average specific flow is $q_{average}$ = 0.65 (with a march width of 2.6 m) and $q_{average}$ = 0.8 person/s/m (with a march width of 4 m). It was found that the Colosseum complies with current standards for on-time evacuation; while modern sports and entertainment arenas are currently designed with additional services, infrastructure, comfort and, in general, high commercialization. The antique arenas are currently being reborn and used for concerts and other public events, so the obtained results have practical significance.

**Keywords:** design; stadiums and arenas; evacuation time; safety; Colosseum; organizing evacuation; computer simulation



## 1. Introduction

Sports and entertainment stadiums with a large number of people are high-risk facilities. A source of hazard is the simultaneous presence of thousands of people in them. The greatest danger is posed by the operating conditions with the simultaneous targeted pedestrian movement, including the stadium outflow after events and the emergency evacuation, e.g., during a fire case. An important role belongs to the space-planning decisions of the structure: the size, configuration, and number of evacuation routes to leave the stands and the building in relation to the arena's capacity.

Computer simulation is widely used to analyze the infrastructure during public events and the operation of space-planning decisions [1,2]. For example, in Ronchi et al. [3], three scenarios of the evacuation of music festival locations with a capacity of 65 thousand people were explored. Simulations of pedestrian movement in the stands are considered in Was et al. [4], Wagner et al. [5], and Zhang et al. [6]. Simulation of the evacuation from the Wuhan Sports Center Stadium (one of the largest gymnasiums in China) was considered in Zong et al. [7]. In Wei et al. [8], the simulation technology of fire spread and evacuation in a large stadium was studied. In Kirik et al. [9,10], the authors presented the effects of different stadium features on evacuation times and densities, which were found using simulation.

Computer simulation provide numerical results for analyzing the object, verifying various hypotheses and obtaining reliable conclusions based on the simulation. Many works have been published aimed at the accurate reproduction of cultural heritage objects using digital technology. For example, [11] describes a digital 3D reconstruction of Sinhaya, a X–XIIth century Muslim suburb in the city of Zaragoza, as a result of which its exact models are obtained. The visualization is based on archaeological evidence from excavations and accurate historical documents. Digital reconstruction has helped to preserve some of the city's cultural heritage. In Papagiannakis et al. [12], a digital visualization of the 16th century Mosque of Hagia Sophia is presented in order to introduce virtual cultural heritage objects into an educational and recreational program. In Heigeas et al. [13], a modeling process is presented to produce a realistic crowd simulation in the ancient Greek agora of Argos. This paper considers the movement of crowds submitting to a common flow in a constrained environment. In Cain et al. [14], a study aimed to create a real-time interactive scenario in the ancient Roman Odeon in Aphrodisias based on historical sources is described. The results of the work present the development of the main scenarios of crowd movement.

Buildings with mass gatherings are not only the heritage of the contemporary world but similar arenas were also built in ancient times. The Roman Colosseum, which is the most famous structure of antiquity and was commissioned in 80 A.D., was built for gladiatorial games, mock naval battles, animal hunts, and the execution of criminals. The Colosseum is the largest amphitheater ever built, with an estimated capacity of 40,000 to 50,000 people [15,16]. The Colosseum was built of travertine stone, tuff, and brick, with marble as a facing material [17]. These materials are not combustible, but there was a fuel load in the building: on the upper tier, there were wooden masts and yards with sunshades on them; at the bottom (basement, under-stand galleries), there were wooden cages for animals, hay, fabrics, stretchers, baskets, etc. An open fire was used for lighting.

In Tan et al. [18] and Hernández [19], a goal was to reconstruct the Colosseum building using a computer model, and in Napolitano et al. [20], a model was created. A computer simulation of masonry in the stone structures of the Colosseum was used. In Croci [21], the weakness of the building concerning earthquakes is outlined. The influence of the space-planning decisions of the Colosseum on the evacuation time is partially considered in Gravit et al. [22]. According to [23], the Colosseum has such space-planning decisions that it is possible to fill and leave the amphitheater within a few minutes. It is estimated that due to the efficiency of the stairs, a full audience is able to leave the Colosseum in three minutes, which is disputed by the authors in [24]. This paper presents a digital reconstruction of the Colosseum to simulate crowd movement, which results in the identification of potential bottlenecks preventing rapid (timely) evacuation. As an effective evacuation scenario for the Colosseum, in [25], a comparison was made with one of the stadiums of modern times, the Beijing National Stadium ("Bird's Nest") built for the 2008 Olympic Games, on the TV show Time Scanners (on the National Geographic Channel). The experiment focuses on the ability of both stadiums to evacuate visitors in the shortest possible time: 1/8th of the Colosseum and the Bird's Nest Stadium were created to reproduce the stadium bowls, corridors, and stairs within seating. The experiment was conducted with two control measurements: full evacuation of people from the stands and full evacuation from the stadiums. According to the results of the first part of the experiment, it took 4 min for spectators to leave the stands in Beijing Stadium, while in the Colosseum during this time people were still in the stands, which means that the design of the exits and stands in the Bird's Nest Stadium is better in evacuation compared to the Colosseum. In the second part of the experiment, as the flow continues, the crowd density in the ancient amphitheater begins to decrease over time due to the configuration and width of the stairs, whereas in the modern stadium the flow begins to slow down and accumulate due to the integrated infrastructure. As a result, the last person left the Colosseum in 12 min 44 s and the last person left the Bird's Nest Stadium in 12 min 57 s. Thus, studying ancient objects and comparing them with modern objects is an actual task.

The purpose of this study was to simulate the space-planning decisions of two sports and entertainment arenas of different epochs: the Roman Colosseum (Italy) and Gazprom Arena (Russia) for a comparative analysis of the organization of pedestrian evacuation, with regard to the geometric characteristics of the stairs affecting the carrying capacity. The following tasks are set to achieve this purpose: to analyze the space-planning decisions of the considered arenas and on their basis to develop 3D models; to calculate and compare the movement of people on the stairs; to determine evacuation time, fields intensity of movement, and density of people.

## 2. Materials and Methods

### 2.1. Evacuation Modelling

In case of fire, the facility's smoke protection system plays a decisive role in ensuring safe evacuation conditions. The safe conditions are currently defined by the inequality (1):

$$t_{evac} < \alpha \, t_{block} \tag{1}$$

where $t_{evac}$ is the time of the end of evacuation from the building area, $t_{block}$ is the time of reaching the critical value by any dangerous fire factors, and $0 < a < 1$ is a safety factor (for example, it equals 0.8 in Russia) [26].

The quantitative characteristics were obtained using the computer simulation of the movement of people (evacuation) in the Sigma FS (Russia) software package for the advanced fire and evacuation simulation [27,28]. The software was used to check the designs and organize pedestrian areas for the 2018 FIFA World Cup and the 29th Winter Universiade athletics facilities [9,29].

An individual flow model was built to simulate the evacuation. The model suggests the calculation of each person's position, including the positions of other people and obstacles on the plane, and allows one to specify individual characteristics, including the free movement velocity, projected area, path, and movement start time. The individual flow model is best suited for simulating the pedestrian traffic on facilities with stands.

At each time instant $t$, the position of each person is determined by the previous coordinate by the formula (2):

$$\vec{x}_i(t) = \vec{x}_i(t - \Delta t) + \vec{v}_i(t)\Delta t, \; i = \overline{1, N}, \tag{2}$$

where $\vec{x}_i(t - \Delta t)$ denotes the particle's position at the previous time step; $\vec{v}_i(t), i = \overline{1, N}$ is the particle's current speed measured in [m/s]; and $\Delta t$ is a time shift equal to 0.25 s.

A person's speed depends on density [30–32]. It is assumed that only conditions in front of the person influence on speed. It is motivated by the front-line effect (that is well pronounced while flow moves in open boundary conditions) in a dense mass of people, which results in the diffusion of the flow.

Thus, only density $F_i(\hat{\alpha})$ in the direction chosen is required to determine the speed. According to [30,33] the current velocity of the particle may be calculated, for example, by formula (3):

$$v_i(t) = \begin{cases} v_i^0 \left(1 - a_l \ln \frac{F_i(\hat{\alpha})}{F^0}\right), & F_i(\hat{\alpha}) > F^0; \\ v_i^0, & F_i(\hat{\alpha}) \leq F^0, \end{cases} \tag{3}$$

where $F^0$ is the limit people density until which free people movement is possible (density does not influence on the speed of people movement); $a_l$ is the factor of people adaptation to current density while moving on $l^{th}$ kind way ($a_1 = 0.295$ is for horizontal way; $a_2 = 0.4$, for downstairs; $a_3 = 0.305$, for upstairs).

An individual flow model was built using the Sigma FS software to simulate the evacuation. The following individual characteristics of people were used in the calculation:

1. The average maximum velocity of a person's free movement was taken to be 1.66 m/s [33];

2.  The fundamental diagram of the relation between the velocity and the current flow density was borrowed from [33] (the assumption that this diagram is fully justified for the Colosseum is based on the analysed data in terms of the limiting flow rate and dynamics);

3.  The person horizontal projection area used was 0.1 m$^2$ [33]. The differences in gender, age, health status, and other indicators were ignored.

Simulation of the movement of each individual and the phenomena peculiar to the flow of people: merger, reshaping (spreading, compaction), the non-simultaneous merging of flows, formation and deformation of congestions, flow around turns, and movement in rooms with a developed internal layout, counter-flows, and intersecting flows are performed.

### 2.2. Description of the Arena Designs—Gazprom Arena Stadium

Gazprom Arena (Russia) is the most visited stadium in Eastern Europe, commissioned at the end of 2016 and hosting the 2018 FIFA World Cup and the 2020 UEFA European Football Championship [34].

According to the technical specifications of the building, a stadium bowl is designed for 68,000 seats, including temporary stands, which can be installed on the third- and sixth-floor stylobates. When the field is involved, the stadium capacity in the concert regime is increased to 80,000 people. The bowl consists of two (lower and upper) tiers. The height difference between the lower tier rows is almost 12 m. There are exits (safety hatches) to the second floor and to the inner stylobate located on the third floor (the attitude is +14.550). The lower bowl is almost symmetrical relative to the minor axis of the field. The height difference between the upper tier rows is almost 20 m. There are exits (safety hatches) to the fifth (+25,200) and sixth (+32,850) floors. The upper bowl can be considered symmetrical with respect to both axes. Figure 1 shows a north-eastern view of the Gazprom Arena and a 3D model of the Gazprom Arena (north-eastern view), built with Sigma FS software.

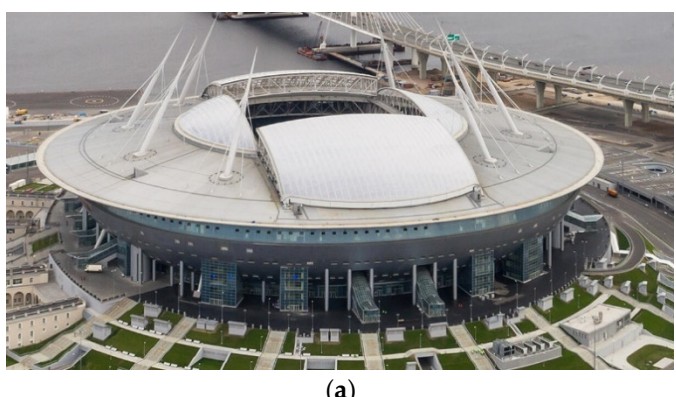

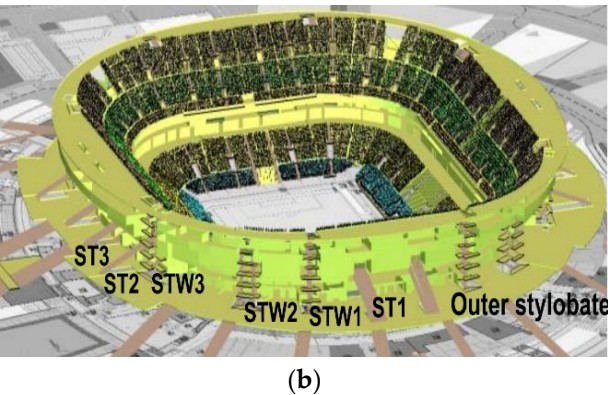

(**a**)             (**b**)

**Figure 1.** (**a**) Northeast side view of the Gazprom Arena stadium; (**b**) 3D model of the Gazprom Arena stadium (north-eastern view) built in the Sigma FS software.

The emergency exits from the building for the lower bowl audience are located mainly on the third floor (only the eastern-sector audience can exit outside directly from the second floor below the third-floor outer stylobate). The exit outside from the upper bowl is also located at the third-floor level. For this purpose, there are stairs accessed from the fifth and sixth floors. The audience members go out to the third floor outer stylobate from the stairs outside. There are 12 such access stairs along the stadium perimeter. In Figure 1, there are marching staircases STW with a number corresponding to the north-eastern quarter of the arena and running from the sixth floor. In addition, straight (no marches) stairs ST with a number are available to descend from the fifth floor directly to the third floor of the stylobate. The audience members descend from the third-floor stylobate to the grade.

In this study, the evacuation of the Gazprom Arena was considered from the upper bowl of the investigated quadrant. We assumed that the exit from the upper bowl would be

the exit to the stylobate, located on the third floor, due to the space-planning similarity and comparable capacity of the Gazprom Arena and the Colosseum. Figure 2a shows the plan of the upper bowl of the north-eastern part of the Gazprom Arena, specifying the number of people in the stands. The numbers of people going to the fifth and sixth floors are shown, and the stairs that can be used to descend are indicated (STW1 and STW3). Figure 2b shows a plan of the fifth-floor under-stand space. Stairs accessible from the fifth floor to the third floor (STW1, STW2, and STW3) and straight descents directly to the third floor outer stylobate (ST1, ST2, and ST3) are marked. The arrows show the directions of movement from the hatches to the nearest exits from the floor; the numbers of people for whom the corresponding exit is the nearest one are indicated (the total number of people is 4454). The stairs are distributed around the fifth floor fairly uniformly. In this case, the loads on the adjacent stairs differ by a factor of up to 2. The stairs-to-sector ratio is 6/9. Figure 2c presents a plan of the sixth-floor under-stand space. The stairs accessible for descending from the sixth to third floor (STW1, STW2, and STW3) are shown. The arrows show the directions of movement from the hatches to the nearest exits from the floor; the numbers of people for whom the corresponding exit is the nearest one are indicated (the total number of people is 4692). The analysis of the sixth-floor plan shows that the number of stairs in it is twice as small as on the fifth floor, while the number of audience members on the former is greater. The stairs-to-sector ratio is 3/9. The stairs are nonuniformly distributed relative to the hatches, the loads on the stairs differ by a factor of more than 2, and the minimum load is twice as high as that on the fifth floor.

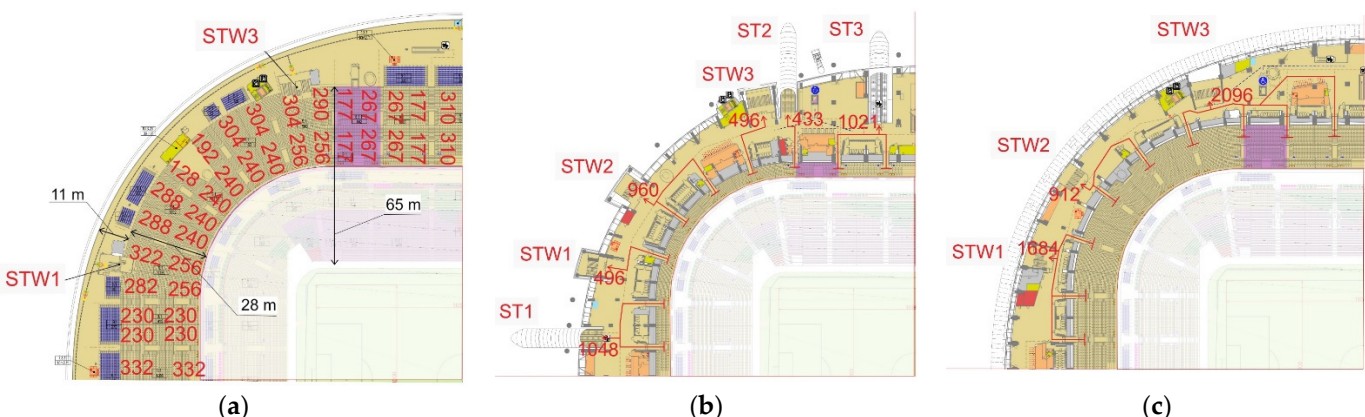

(a) (b) (c)

**Figure 2.** (**a**) Plan of the upper bowl of the north-eastern part of the Gazprom Arena and the number of people in the stands; (**b**) plan of the north-eastern part of the Gazprom Arena fifth floor; (**c**) plan of the north-eastern part of the Gazprom Arena sixth floor.

For further analysis, only stairs STW1, STW2, and STW3 are considered, since they are used by people descending from two (fifth and sixth) floors. In addition, the design of stairs STW1 is significantly different from that of stairs STW2 and STW3 (Figure 3). The quantitative data are given in Table 1.

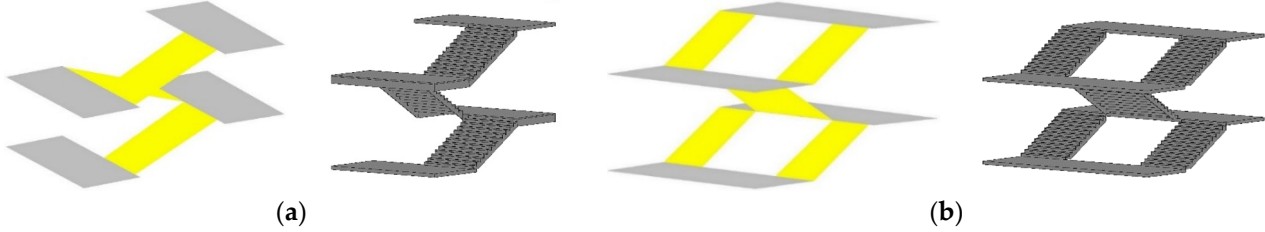

(a) (b)

**Figure 3.** (**a**) Fragment of the stairs STW1 with a stair flight of 2.6 m; (**b**) fragment of the stairs STW2 and STW3 with a stair flight of 4 m (2 + 2 m).

**Table 1.** Loads on stairs STW1, STW2, and STW3 and their geometric dimensions.

| Name of Stairs | Fifth Floor, Persons | Sixth Floor, Persons | Total, Persons | Minimum Width, m | I, Person/m of Width | Stair Length along the Axis of Movement, m |
|---|---|---|---|---|---|---|
| STW1 | 496 | 1684 | 2180 | 2.6 | 838.5 | 63 |
| STW2 | 960 | 912 | 1872 | 4.0 | 468 | 63 |
| STW3 | 496 | 2096 | 2592 | 4.0 | 648 | 63 |
| Total, persons | 1952 | 4692 | 6644 | | | |

The minimum path width for stairs STW1 is 1.5 times less than for the stairs STW2 and STW3, although the number of people evacuating on the stairs STW1 (2180) is comparable to the number evacuating on the stairs STW2 (1872) and STW3 (2592). The ratio between the discharge values for these stairs is the same. Calculating the stairs loading according to the nearest stairs principle, it is clear that the staircase with the lowest discharge value (STW1) on the sixth floor has an almost maximum load: the stairs take half of the northern part of the sixth floor. At the same time, the adjacent stairs STW2 with a discharge value greater by a factor of 1.5 are only accessed for two sectors located directly on the corner. The load on stairs STW1 on the fifth floor is reduced by the presence of exit ST1.

*2.3. Description of the Arena Designs—Colosseum*

There has still been no consensus among historians and architects about an antique amphitheater's design features and appearance. The characteristics that are important for the study and included in the three-dimensional computer model of the building to simulate evacuation and analyze the results obtained are considered. The computer model of the Colosseum is based on Durm's structural scheme [16]. During the simulation, the main attention is paid to the under-stand space, and the stairs for descending from the upper tiers since this part of the building affects the evacuation time the most.

The Colosseum central part is an oval stage surrounded by a flat strip of seats; the ratio between the major and minor axes of the entire building is 1.22. An oval cone with seats is around the arena. It is based on 80 parting walls directed radially inward and interconnected by ring walls and arched rows. Between them, there are a corresponding number of radially directed crossings and staircases; ring galleries stretching along the entire amphitheater between the ring walls and arcades connect walkways and stairs. The exterior galleries of the second and third floors serve as lounges. The gallery height on the floors is 10–11 m.

There are 80 arches along the outer perimeter that form 80 amphitheater entryways (Figure 4). The entrances/exits are located at the ground level (the so-called datum). Therefore, the evacuation can only occur top-down.

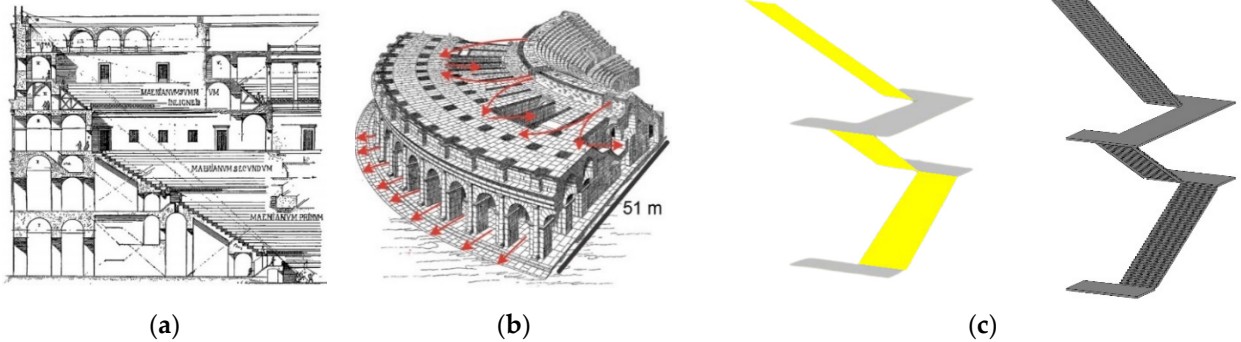

| **(a)** | **(b)** | **(c)** |

**Figure 4.** (**a**) Schematic of the Colosseum architectural design according to Durm's representations [16]; (**b**) schematic view of the Colosseum second floor (Gyuade) [16]; (**c**) fragment of the stairs.

The amphitheater can be conventionally divided into three tiers, each containing under-stand galleries, stands, and walkways to the seats (Figure 5).

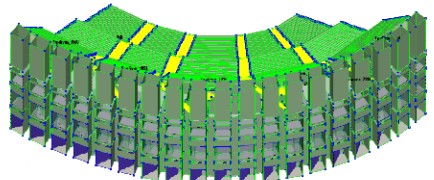
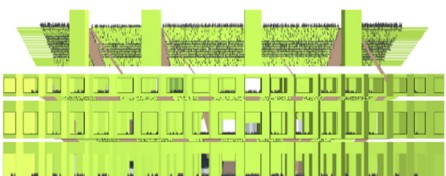
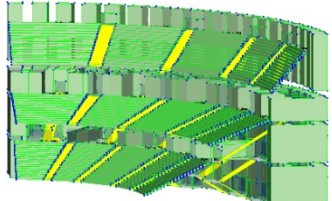

**Figure 5.** Sigma FS software 3D models of the Colosseum.

The model was built assuming that the access to the ground-tier stands was mainly through the second floor; to the second-tier stands, through the third floor; and to the third-tier stands, through the fourth floor (the attitude of the latter is about +40,000). The first two tiers represent sequences of 20 stand rows, and the upper tier contains 16 rows. The data on the maximum arena capacity reported by different authors are inconsistent and vary between 40 and 50 thousand audience members simultaneously [16], so the conventional number of people is 48,000.

The Colosseum has the line-of-sight downstairs on both sides of each exit to the under-stand gallery (Figure 4b). The simulation considered 1/4 of the Colosseum (calculation sector), where 4 stairs are taken to evacuate people, which are located in this sector. The extreme stairs on two opposite sides of the calculation sector take the remainder of the flow for each subsequent sector. The stairs are uniformly distributed along the floor perimeter. The number of stairs is consistent with the number of exits to the under-stand space, i.e., it is equal to the number of tier sectors. The stairs path width along the axis of movement ranges from 2 m for descending from the upper tier of stands to the third floor to 4.5 m in the lower part.

*2.4. Initial Data for the Evacuation Simulation*

To compare the two arenas, a quarter of the Colosseum and a quarter of the Gazprom Arena's upper bowl are considered. This is justified by the symmetry of the Colosseum and the Gazprom Arena upper bowl with respect to both axes; in addition, the buildings have comparable capacities (12,000 and 9500 people, respectively), and the only way to evacuate is down the stairs.

Figures 6 and 7 show the 3D models built for the arenas. The Gazprom Arena computer model was built using modern drawings. The entire stadium was modeled and used not only within the limits of this study.

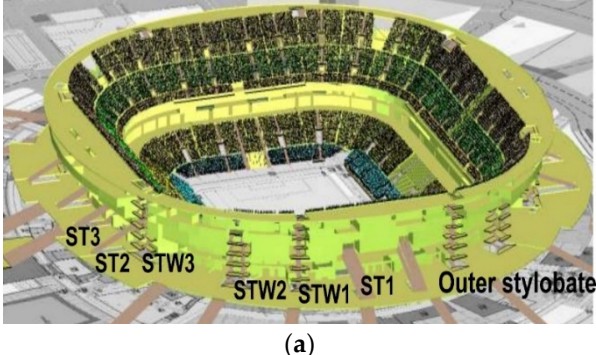
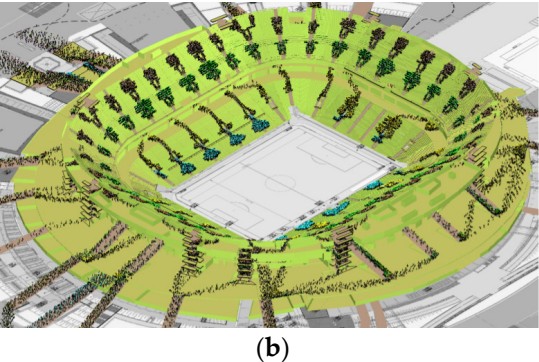

(**a**) (**b**)

**Figure 6.** (**a**) The position of people in the stands of the Gazprom Arena before the evacuation; (**b**) the position of people during evacuation from the Gazprom Arena at the hundredth second from its beginning.

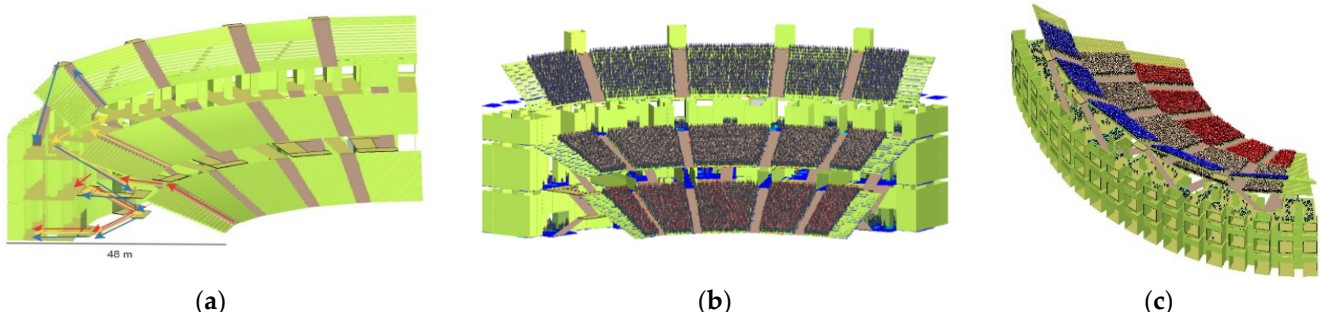

**Figure 7.** (**a**) View from the side of the Colosseum; (**b**) view from the center of the arena; (**c**) view from the front of the building.

The Colosseum computer model is based on Durm's structural scheme [16]. The attention was mainly paid to the under-stand space, and the stairs for descending from the upper tiers since this part of the building affects the evacuation time the most. The arrangement of the stairs for descending from the upper tiers is approximately the same around the perimeter of the arena, so, when building the computer model, the approximate length and width of each unit path down from the upper floors and the number of paths (stairs) are provided.

Many geometrical dimensions of the interior space of the Colosseum were taken at a scale relative to the known dimensions given in the drawings. The descriptions provide limited data on the configuration of the stairs used to descend from the upper tier to the third floor. However, it is known that people from the upper tier merged into the streams of people from the corresponding sectors of the lower tier. Therefore, the stairs for descending from the upper-tier were conditionally restored to ensure the descent of a number of persons significant for further consideration in the general flow to the third floor. Each sector of the stands on each tier has a staircase for descending from the sector to the underlying floor, where people use the nearest stairs to descend further. The model includes 5 sectors. They are secured by 5 access staircases. In order to exclude boundary effects, the dynamics of human movement in the central part was analyzed, i.e., in the three central sectors and the four central staircases. For the same reason, the extreme sectors in the model are only half-filled (Figure 7b).

The computational domain involved the stands, under-stand galleries, and stairs. At the initial instant of time, people were in the stands or in the under-stand space. The evacuation of people from the building was simulated before exiting outside at the first-floor level for the Colosseum and before exiting beyond the exterior perimeter to the stylobate for the Gazprom Arena.

## 3. Results and Discussion

*3.1. Comparative Analysis of the Arenas Using the Numerical Simulation of Human Movement*

Figure 8 shows a fragment of the Colosseum evacuation at the hundredth second from its beginning and mass gathering intensity field on the Colosseum third floor. Figure 9 shows the fields intensity of movement and crowding.

There were 7 calculations (scenarios) for the Gazprom Arena with different staircase loads STW1 and STW2-3. The last two scenarios (6 and 7) are proposed in the absence of flow control on the fifth and sixth floors with an uneven distribution of stairs, which is explained by the use of certain sectors for the needs of different client groups. The data on number $N$ of the persons who passed the stairs, spent time $t$, and flow rate $Q$, determined by formula (4), are given in Table 2.

$$Q = N/t \qquad (4)$$

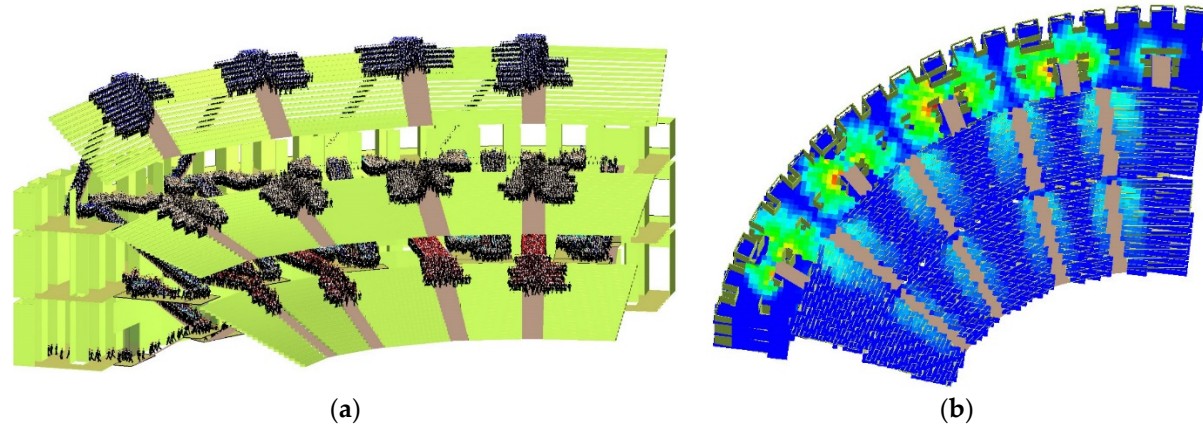

**Figure 8.** (**a**) Fragment of the Colosseum evacuation at the hundredth second from its beginning; (**b**) mass gathering intensity field on the Colosseum third floor.

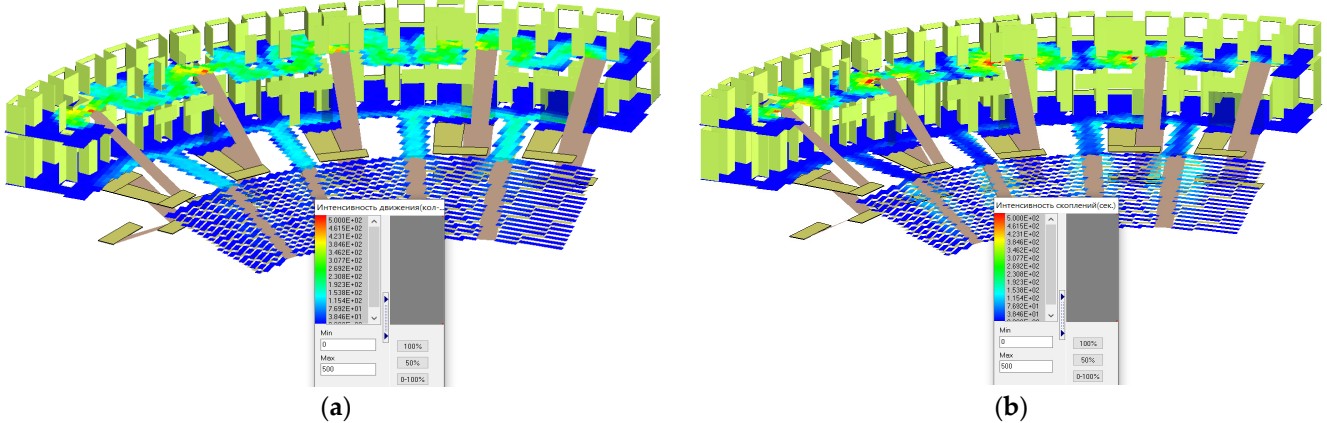

**Figure 9.** (**a**) Field of total traffic intensity in seconds on the second and third floors of the Colosseum; (**b**) intensity field of crowding in seconds on the second and third floors of the Colosseum.

**Table 2.** Numerical characteristics of the Colosseum and Gazprom Arena.

| | Gazprom Arena. the Height Difference Is 18.3 m | | | | | | Colosseum. the Height Difference Is 22 m | | |
| --- | --- | --- | --- | --- | --- | --- | --- | --- | --- |
| | STW1, the Width Is 2.6 m | | | STW2-3, the Width Is 4 m | | | the Width Is 2.8 m | | |
| | $N$ | $t$, s | $Q$, Person/s | $N$ | $t$, s | $Q$, Person/s | $N$ | $t$, s | $Q$, Person/s |
| 1 | 2 | 3 | 4 | 5 | 6 | 7 | 8 | 9 | 10 |
| 1 | 1680 | 990 | 1.7 | 1810 | 520 | 3.5 | 2150 | 705 | 3.1 |
| 2 | 1800 | 1075 | 1.7 | 1970 | 600 | 3.3 | 2405 | 760 | 3.2 |
| 3 | 2030 | 1175 | 1.7 | 1980 | 640 | 3.1 | 2480 | 740 | 3.4 |
| 4 | 2380 | 1400 | 1.7 | 2010 | 660 | 3.0 | 2720 | 840 | 3.2 |
| 5 | 2850 | 1570 | 1.8 | 2150 | 640 | 3.4 | | | |
| 6 | 3580 | 2080 | 1.7 | 2380 | 725 | 3.3 | | | |
| 7 | 3670 | 2025 | 1.8 | 4410 | 1290 | 3.4 | | | |
| 8 | Mean | | 1.7 | Mean | | 3.3 | Mean | | 3.2 |

According to Table 2, rows 1–4 do not account the remaining number of evacuees in the Colosseum, which are on the extreme staircases on two opposite sides of the calculation sector.

The data are given in columns 4 and 7 confirm the expected difference (by a factor of about 2) between the flow intensity estimates for stairs STW1 and STW2-3 because of the similar difference between the path widths. At similar numbers of persons, the evacuation time for stairs STW1 is twice as long as for stairs STW2-3.

It is worth noting that the capacity of the Colosseum stairs is comparable with that of stairs STW2-3 in the Gazprom Arena. Meanwhile, the staircase width in the Colosseum is smaller by a factor of ∼1.5. The construction of the stairs causes this effect. In the Colosseum, the height difference between the third and first floors is 22 m; in the Gazprom Arena, the height difference between the investigated sixth and third floors is 18.3 m. These values can be considered similar. The structure of the Gazprom Arena stairs was accurately reconstructed in the computer model. The main important features are that all the stairs connecting the upper floors are outside the bowl. There are eight 180° turns between the sixth and third floors (a stair flight has an average height difference of 2.1 m and an average slope of 30°; the flight widths are given in Table 1).

The evacuation time for the considered part of the Gazprom Arena ranges from 520 to 2080 seconds and depends on the load of the stairs and can be regulated by the organisation of the human flow. The evacuation time from the Colosseum is 14.5 min, taken as the sum of the maximum time to leave the stairs of the sector (840 s) and the additional time to exit from the structure (30 s).

In order to assess the results obtained for the Colosseum, it should be noted that the interior space (in particular the staircases) has been reconstructed approximately. However, the space-planning decisions of the Colosseum floors, which is still accessible for research, and the data on the under-stand space structure and the axes lengths in the plan allow to consider the geometry of the Colosseum vertical connections used in the model to be sufficient for this study. In particular, the descent from the third to second floor was reconstructed as straight (without turns, its length is 21.5 m); it occupies the under-stand space of the second tier. The stair flights going down from the floors are codirected; to reach the next flight, one needs to make two 180° turns. There is one 180° turn between the second and first floors, and there are three turns to make in total when descending from the third and first floors; the average flight slope is 30°.

Table 3 generalizes the numerical characteristics of the investigated stairs for the two arenas. The Colosseum stairs are characterized by the highest specific flow (column 5). With conditionally the same length, slope, and height difference parameters, this fact is ensured by the layout of the Colosseum stairs, specifically, by the number of turns (column 6), which is twice as small as in the Gazprom Arena stadium. The result obtained is consistent with the data of a full-scale experiment [35], in which the movement downstairs in a nine-storied building was examined; there were 180° turns on the stairs, and the specific flow decreased with a decrease in the floor (and an increase in the number of turns).

**Table 3.** Summary table with the numerical characteristics of the Colosseum and Gazprom Arena stairs.

| | Stairs | Width, m | $Q_{av}$, Person/s | $q_{av}$, Person/s/m | Number of 180° Turns | Height Difference, m | Length, m | Slope, Deg |
|---|---|---|---|---|---|---|---|---|
| 1 | 2 | 3 | 4 | 5 | 6 | 7 | 8 | 9 |
| 1 | Gazprom Arena, STW1 | 2.6 | 1.7 | 0.65 | 8 | 18.3 | 63 | 30 |
| 2 | Gazprom Arena, STW2-3 | 4 | 3.3 | 0.8 | 8 | 18.3 | 63 | 30 |
| 3 | Colosseum | 2.8 | 3.2 | 1.14 | 3 | 22 | 63 | 30 |

Thus, under other conditions, which can be assumed to be the same or slightly different for the investigated arenas, a key characteristic that determines the building evacuation rate was found to be the geometric characteristic of the stairs determining the number of 180° turns. The relation between the specific flow and the number of turns is nonlinear. In addition, as can be seen from rows 1 and 2 of column 5, the configuration of the stairs (Figure 3) also affects the flow rate. Table 4 shows the main geometric characteristics of the Colosseum and Gazprom Arena stairs.

**Table 4.** Summary table with the numerical characteristics of the Colosseum and Gazprom Arena stairs.

| | Characteristic | Colosseum | Gazprom Arena |
|---|---|---|---|
| 1 | Minimum downstairs flow rate, person/m/s | 1.14 | 0.65 |
| 2 | Number of 180° turns per stairs | 3 | 8 |
| 3 | Average mass gathering time, s | 360 | 900 |
| 4 | Evacuation control (routing) to balance the load on the stairs and reduce the time of mass gathering in front of the stairs | not required | required |
| 5 | Stage-by-stage evacuation | not required | required |
| 6 | Fencing the escape routes from the main space | no | yes |
| 7 | Protection against the dangerous fire factors | Stairs configuration ensuring the high velocity of movement | Fenced-off staircases protected from the spread of the dangerous fire factors |
| 8 | Free path to the adjacent hatch along the stand | yes | no |
| 9 | Availability of a staircase for each stand (stairs/stand) | 1/1 | 2/3 (fifth floor); 1/3 (sixth floor) |

*3.2. Discussion*

The most reliable smoke protection methods are the use of optimal space-planning decisions of buildings and structures.

The Colosseum is an open structure, where, in case of fire, there are almost no obstacles for spreading the dangerous fire factors, including, first of all, smoke, in the under-stand space; therefore, the speed of evacuation from the building is a decisive factor. The high velocity of movement of people from the upper tiers is ensured by the escape routes maximally straightened using the optimal configuration of the stairs and providing each stand with its own downstairs and own exit from the building. In the Colosseum, the people gathering places with a density of 6 [person/m$^2$] and higher are the exits to the downstairs on the third floor (Figure 8b), since the capacity of these stairs is lower than the intensity of flows from the second and third tiers. Therefore, the time of mass gathering on the third floor can be minimized by the phased evacuation.

In the Gazprom Arena, the under-stand space is fenced off from the environment (in contrast to the bowl, which, in general, can be considered open). The smoke protection by design is implemented via walling off the staircases and making them smoke-free. The availability of downstairs in the Gazprom Arena upper bowl ranges within 1/3–2/3 on different floors. This leads to the discrepancy between the intensities of the suitable flow and the discharge values of the doors on the staircase and causes the long-term (up to 900 s) mass gathering (Table 4). The problem can be solved by organizing the phased evacuation. To enhance the efficiency of using the vertical lines, it is necessary to control the human flows on the fifth floor in order to relieve stairs STW1-likewise, which take a significant load in the south and north sectors of the sixth floor.

**4. Conclusions**

Currently, ancient arenas are being reborn: They are used for concerts and other public events, so research on the calculation of evacuation times from such structures is relevant and meaningful. In addition, the Colosseum is the prototype of most modern sports facilities in the present (Fisht Stadium (Sochi, Russia) and the Bird's Nest Stadium (Beijing, China)).

The evacuation process from Colosseum (Italy, Rome) and the Gazprom Arena (Russia, St. Petersburg) is investigated using pedestrian dynamics simulation. The effect of the design of evacuation paths on evacuation time(s) is studied, and the need to optimally organize evacuation (assist in loading stairs) is found.

According to results of investigation, the Colosseum design seems advantageous over the Gazprom Arena. The most significant difference is the higher stability and weak need

of the evacuation process in the control factors. The key issue is the uniform distribution of vertical communication ways around the perimeter of the arena, the balance of the capacity of the escape routes and the intensity of the flow, which is also achieved due to the geometric features of the escape routes—the straighter the path, the higher the speed of movement.

The greatest intensity of human flows in the Colosseum is recorded on the third floor, because spectators are flocking here from the two tiers (second and third). There are also the longest crowds (the average duration is 200–250 s).

In the Colosseum, the high speed of movement of people from the tiers is realized by maximally straightened evacuation routes (staircase configuration and provision of each tribune with its own staircase). The stairwells at the Gazprom Arena are walled off and separated from the general volume of the stadium bowl, in particular, from the under-stands premises. The availability of staircases for the upper tier stands at the Gazprom Arena varies between 1/3 and 2/3 of the floors. The key characteristic determining the building's evacuation rate is the number of 180° turns.

According to the simulation results, the evacuation from the upper bowl of the Gazprom Arena to the stylobate of the 3rd floor ranges from 9 to 35 minutes. The evacuation depends on the location and load of the stairs, which is uneven and can be regulated by organising the flow of people. The evacuation from the Colosseum is 14.5 minutes, as the stairs are designed to be evenly loaded and symmetrically arranged. When the flow is organised appropriately in the Gazprom Arena, the structures have similar evacuation times. With an average march width of 2.8 m, the average specific flow $q_{average}$ = 1.14 person/s/m (in the Colosseum), and 0.65 and 0.8 person/s/m (in the Gazprom Arena on the STW1 and STW2-3 types of stairs, respectively).

The Colosseum is designed with large, long staircases using the principle of Vomitoria, which means eruption. This study proved the effectiveness of the stairs used in the Colosseum. In the construction of a structure in order to ensure the shortest possible evacuation time, this solution is the most effective. According to this study, the Colosseum complies with current standards for timely evacuation and can be operated as a modern sports and entertainment facility and host public events.

The main difference between modern sports and entertainment arenas is that they are designed with additional services, infrastructure, comfort and, in general, high commercialization, which has an impact on evacuation times and requires additional resources for the application of organizational management of the flow of the people.

**Author Contributions:** Conceptualization, M.G.; software, E.K.; investigation, T.V.; formal analysis, E.S.; data curation, D.S. All authors have read and agreed to the published version of the manuscript.

**Funding:** The research is partially funded by the Ministry of Science and Higher Education of the Russian Federation under the strategic academic leadership program "Priority 2030" (Agreement 075-15-2021-1333 dated 30 September 2021).

**Institutional Review Board Statement:** Not applicable.

**Informed Consent Statement:** Not applicable.

**Acknowledgments:** The authors would like to thank Nikolai Ivanovich Vatin, Peter the Great St. Petersburg Polytechnic University, St. Petersburg, Russia, for valuable and profound comments.

**Conflicts of Interest:** The authors declare no conflict of interest.

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
