# Peer review of "Simulation of Evacuation from Stadiums and Entertainment Arenas of Different Epochs on the Example of the Roman Colosseum and the Gazprom Arena"

_fire, doi:10.3390/fire5010020_

Round 1

Reviewer 1 Report

Title of manuscript: Simulation of evacuation from stadiums and entertainment arenas of different epochs on the example of the Roman Colosseum and the Gazprom Arena.

Journal: Fire (ISSN 2571-6255) 

Manuscript ID: fire-1551148

Special Issue: Performance-Based Design in Structural Fire Engineering

Jan 2022

The article entitled "Simulation of evacuation from stadiums and entertainment arenas of different epochs on the example of the Roman Colosseum and the Gazprom Arena" is an interesting study in the field of fire safety, especially in the area of evacuation from large sports or entertainment venues. The topic fits well into the scope of the Fire. I found it to be an ingenious, highly original approach of extraordinary importance and significance in the field of fire safety, fire engineering and the development of self-protection plans.

I found the article very well organised and structured. The state of the art is adequate, the methodology is very precise and the results obtained are in accordance with the methodology used.

As for the conclusions, I think they are appropriate, although I would advise making clear the limitations encountered in carrying out the study and the projection of future lines of research recommended by the authors.

As a very minor improvement I would advise:

- The referencing style used for citations in the manuscript for the most part does follow the standard referencing style recommended by this journal. However, I note some elements for improvement, to be exposed:

Line 50. The phrase "in [6-8]" I recommend changing it to: "in Was et al [6], Wagner et al [7] and Zhang et al [8]".

Line 50 The sentence "In [9], a system for" should be changed to "In Wagoum and Seyfried [9], a system for".

In short, I would like to congratulate them for their work and, of course, my verdict as reviewer of this manuscript is that it is accepted.

Reviewer 2 Report

Dear authors,

Thanks for your contribution to Fire.

With major revisions of the manuscript, it might be accepted.

The opinions are set out below:

  1. Please prepare it in a concise manner and reduce redundancy.
  2. The spelling and writing of this manuscript must be accurate.
  3. The abstract is unclear. Please clarify.
  4. Some abbreviations have not been included in the full definition, e.g. What is STW?
  5. What is the Reliability of Sigma FS software? Do you need verification and validation?
  6. Which mesh or grid is used for the simulation?
  7. As described in Figure 1, it is difficult to get the idea.
  8. How will the results of the simulation of various buildings be applied in future projects?

Sincerely yours,

Reviewer 3 Report

This study used an egress modeling tool and compares the egress phenomena of two separate buildings. Unfortunately, I don't see any intellectual contribution of this study to scholarship. It looks like a report of two modeling outcomes and critical findings are not provided. 

Reviewer 4 Report

I have been looking/reviewing this paper from "Buildings" to "Fires". I have observed much improvement and am happy to see how it has evolved through time. For example the following modifications have already been made:

"The abstract is too short. It only explains the title." i believe its DONE

"The motivation paragraph in the introduction is missing. I mean there is a plethora of evacuation simulations already done. What is novel about this? Maybe, a comparison between ancient and modern structures as written in line 22-23. But it needs more explanation and related work. Related work is not there and that relates to weakness in the motivation." I STILL THINK THAT THIS ISSUE HAS TO BE ADDRESSED

"The design and simulation section is impressive. However, the conclusion that " Colosseum design seems advantageous over the Gazprom Arena" means that we have progressed in the wrong direction in buildings (at least in terms of evacuation). I am sure that more qualitative results could have been extracted from the simulation results, followed by some useful recommendations. Why would a reader be interested in knowing that ancient structures were better than modern ones? Yes, there must be some features that are different, but you need to mention them explicitly. Also, how the modern structure could have performed better qualitatively (preferably evaluated based on simulation on modified design)." I THINK THIS ISSUE IS WELL ADDRESSED NOW.

Lastly, the English and grammar are up to mark now. I am again convinced that such studies which are different and take past to investigate, must get a place in journals so that researchers can enjoy them. 

Round 2

Reviewer 2 Report

Congratulations.

The manuscript might be accepted.

Reviewer 3 Report

I still think this paper does not provide any scientific findings. I don't see the value of comparing two different buildings? If they had started with some specific features such as different stair shapes will result in different efficiency of evacuation, I would understand the purpose of the study, but this paper seems to compare an apple to an orange.